# Can Adipose Tissue Influence the Evaluation of Thermographic Images in Adolescents?

**DOI:** 10.3390/ijerph20054405

**Published:** 2023-03-01

**Authors:** Hamilton H. T. Reis, Ciro J. Brito, Manuel Sillero-Quintana, Alisson G. Silva, Ismael Fernández-Cuevas, Matheus S. Cerqueira, Francisco Z. Werneck, João C. B. Marins

**Affiliations:** 1Departamento de Educação Física, Universidade Federal de Viçosa, Viçosa 36570-900, Brazil; 2Departamento de Educação Física, Universidade Federal de Juiz de Fora, Governador Valadares 35010-180, Brazil; 3Faculty of Physical Activity and Sports Sciences (INEF), Universidad Politécnica de Madrid, 28040 Madrid, Spain; 4Departamento de Educação Física, Escola Preparatória de Cadetes do Ar, Barbacena 36205-058, Brazil; 5Instituto Federal de Educação, Ciência e Tecnologia do Sudeste de Minas Gerais, Rio Pomba 36180-000, Brazil; 6Departamento de Educação Física, Universidade Federal de Ouro Preto, Ouro Preto 35400-000, Brazil

**Keywords:** temperature mapping, body composition, adiposity, adolescents

## Abstract

Infrared thermography (IRT) is a technology easy to use for clinical purposes as a pre-diagnostic tool for many health conditions. However, the analysis process of a thermographic image needs to be meticulous to make an appropriate decision. The adipose tissue is considered a potential influence factor in the skin temperature (Tsk) values obtained by IRT. This study aimed to verify the influence of body fat percentage (%BF) on Tsk measured by IRT in male adolescents. A total of 100 adolescents (16.79 ± 0.97 years old and body mass index of 18.41 ± 2.32 kg/m²) was divided into two groups through the results of a dual-energy X-ray absorptiometry analysis: obese (*n* = 50, %BF 30.21 ± 3.79) and non-obese (*n* = 50, %BF 11.33 ± 3.08). Thermograms were obtained by a FLIR T420 infrared camera and analyzed by ThermoHuman^®^ software version 2.12, subdividing the body into seven regions of interest (ROI). The results showed that obese adolescents presented lower mean Tsk values than the non-obese for all ROIs (*p* < 0.05), with emphasis on the global Tsk (0.91 °C) and anterior (1.28 °C) and posterior trunk (1.18 °C), with “very large” effect size values. A negative correlation was observed in all the ROI (*p* < 0.01), mainly in the anterior (r = −0.71, *p* < 0.001) and posterior trunk (r = −0.65, *p* < 0.001). Tables of thermal normality were proposed for different ROIs according to the classification of obesity. In conclusion, the %BF affects the registered Tsk values in male Brazilian adolescents assessed by IRT.

## 1. Introduction

Skin blood flow has been studied for many years, especially for its important role in human thermoregulation. The physiology and vascular anatomy of the skin create a typical pattern of temperature distribution, which must remain within a certain distribution range to be considered healthy. When temperature values deviate from this standard considered ideal, this can be a sign of some kind of illness.

Infrared thermography (IRT) is a non-invasive, radiation-free, and easy-to-apply technology particularly suitable to precisely map the skin temperature (Tsk) through the analysis of thermographic images, which is frequently used for clinical purposes as an auxiliary tool in the process of diagnosis of diseases [1,2,3] and the prevention and rehabilitation of injuries [4,5,6,7]. The procedure is performed using a thermographic camera with a sensor responsible for capturing the heat radiated from the skin’s surface and transforming it into a temperature scale. The camera sensor is positioned close to the evaluated and provides a real-time representation of the Tsk distribution pattern in high resolution.

To obtain a quality thermographic image, the acquisition process must follow specific guidelines, such as the suggested by Moreira et al. [8], and observe several factors that may influence image evaluation. This is suggested in the review written by Fernández-Cuevas et al. [9], which presents studies that indicate that technical, environmental, and individual internal and external factors can influence the analysis by IRT, which is relevant for medical diagnosis purposes or for understanding human thermoregulation processes.

One of the two main factors to be observed, including a group of internal factors, is related to body composition [9]. Body fat has a lower level of thermal conductivity than other tissues involved in the thermoregulation process [10], acting as a “body thermal insulator” [11] since it acts as thermal resistance, making the process of heat conduction of the body more difficult to the internal region of the body compared to peripheral regions (e.g., skin) by 40% to 50% [10] and being able to influence the Tsk of the area where it is more concentrated [12]. Adipose tissue has lower thermal conductivity values than muscle tissue [13], dermis [13], and epidermis [14]. Furthermore, obesity is associated with increased inflammatory cytokines TNF-a or IL-6 to perivascular adipose tissue around healthy blood vessels, which free radical scavengers or cytokine antagonists can block, directly affecting the mechanisms of skin vasodilation and vasoconstriction [15,16].

Some studies have investigated whether the amount of body fat can interfere with the Tsk assessed by IRT in the population of men [17,18,19,20] and women [17,18,21,22,23] and, in general, observed that individuals with a more significant amount of fat presented lower Tsk values in body regions of interest (ROI) such as the trunk, arms, and legs. This factor should be considered during the evaluation of thermal images for a more precise assessment of the results.

Given the need for more precise knowledge on this subject, since the few existing studies are restricted to the adult population [17,18,19,20,21,22,23], and given that the use of IRT is more frequently used in clinical settings, investigating the influence of this characteristic on other age groups seems crucial for increasing the thermal image evaluation capacity of professionals working with IRT.

Thus, the objective of this study was to verify the influence of body fat on the Tsk values of male adolescents and to provide tables of thermal normality that help in the process of evaluating thermographic images and subsequent diagnosis of possible diseases or sports injuries or in helping the physical rehabilitation process. It is hypothesized that %BF will present a negative correlation with Tsk values and that participants with higher amounts of body fat will have lower Tsk values pattern in the regions of the trunk, arms, and lower limbs.

## 2. Materials and Methods

### 2.1. Participants

After evaluating 216 male high school students from public and private schools in a city in the interior of Brazil, we included 100 participants in the study. This amount was based on the total number of participants considered obese after the initial assessment. Thus, we intentionally selected the 50 individuals considered obese (16.83 ± 0.93 years, 78.94 ± 10.08 kg, 1.76 ± 0.07 m height, and a body mass index of 25.63 ± 2.96 kg /m^2^), and to perform a statistical evaluation with the same number of non-obese participants, we randomly selected, among the remaining 166 evaluated, 50 non-obese individuals (16.75 ± 1.01 years, 56.49 ± 8.51 kg, 1.75 ± 0.07 m height, and a body mass index of 18.46 ± 2.50 kg/m^2^). The final characteristics of the sample were 16.79 ± 0.97 years, 67.71 ± 14.61 kg of body weight, 1.75 ± 0.07 m height, and a body mass index of 18.41 ± 2.32 kg/m^2^. As a characterization criterion for individuals with or without obesity, we used the classification proposed by Williams et al. [24] specifically for teenagers. The randomization process of the 166 evaluations was carried out using the website https://www.randomizer.org/ (accessed on 19 December 2022).

As inclusion criteria, we selected male individuals who were apparently healthy, without apparent motor or intellectual deficiency, and aged between 14 and 19. Those excluded from the research were those without a signed informed consent or presenting some of the following exclusion criteria: smoking; history of kidney problems, musculoskeletal injury in the last two months, skin burns, or symptoms of pain in some body region; or sleep disturbances or fever over the previous seven days, physiotherapy or dermatological treatments with creams in the last two days, ointments or lotions for local use in the last two days, consumption of medication affecting Tsk (i.e., anti-inflammatory, antipyretic, or diuretics), or any dietary supplement with potential interference with water homeostasis or body temperature in the last two weeks. In addition, participants could not perform resistance training.

The study was approved according to ethical criteria for research involving human beings by the Ethics Committee of the local Institution under the registration number CAAE 40934275729. After explaining the characteristics and study objective, all the participants (or their person in charge in case of been under 18 years old) voluntarily signed the written consent before participating in the study.

### 2.2. Procedures

#### 2.2.1. Anthropometric Assessment of the Body Fat Percentage (%BF)

All the anthropometric variables were collected by trained professionals with level II certification from the International Society for the Advancement of Kinanthropometry (ISAK) [25]. Initially, height was measured using a portable stadiometer (Cescorf^®^, Porto Alegre, Brazil) with a precision of 1 mm and body mass with a digital balance (Welmy w 200/5, Brazil) with a precision of 0.1 kg. The %BF was determined by dual-energy X-ray absorptiometry (DXA) by a single technician duly qualified for this function, using a GE Healthcare^®^ densitometer, Lunar Prodigy Advance DXA System (software version: 13.31), which provides the values of total and segmented fatness (i.e., trunk, arms, and lower limbs). The equipment was calibrated daily according to the manufacturer’s specifications to guarantee the quality of the measurements.

#### 2.2.2. Thermography Assessment

The thermographic image collection protocol was carried out following what was established by Moreira et al. [8], carefully observing all the factors that need to be considered to obtain a quality image.

Four thermographic images from the upper and lower body (see Figure 1), in the anterior and posterior positions, were registered from each subject using a T420 infrared camera (FLIR^®^, Stockholm, Sweden) located perpendicularly to the center of the recorded body areas. The imager had an accuracy of 2%, a spectral band of 7.5–13 µm, 60 Hz rate, automatic focus, and a resolution of 320 × 240 pixels and could detect temperature variations ≤ 0.05 °C. It was connected at least 30 min before all the evaluations to allow the stabilization of its thermal sensor, setting the emissivity at 0.98. During data collection, ambient temperature (21.3 ± 0.7 °C) and humidity (55.3 ± 2.2%) were controlled according to specific recommendations for this type of evaluation [8,9] and monitored through a portable meteorological station (Instrutherm^®^, THAL-300, São Paulo, Brazil). After stabilizing the temperature and humidity values in the room, the subjects remained standing, wore only slippers and shorts, and avoided any contact with surfaces or scratches for 10 min [26] before the thermographic images were captured. All the thermograms were obtained in the morning to reduce the influence of circadian rhythm on the results [27,28]. The thermal imager was positioned perpendicular to the ground [8] and at a distance allowing the subject to fit into the avatar generated by the software used for analysis so that all ROIs could be satisfactorily evaluated, as shown in Figure 1. After 10 min, following the methodology of Yasuoka et al. [29], they were asked to report the thermal sensation (TS) on a 9-point scale (+4, very hot; +3, hot; +2, warm; +1, slightly warm; 0, neutral; −1, slightly cool; −2, cool; −3, cold; −4 very cold) and the comfort sensation (CS) on a 7-point scale (+3, very comfortable; +2, comfortable; +1, slightly comfortable; 0, neutral; −1, slightly uncomfortable; −2, uncomfortable; −3, very uncomfortable).

The thermograms were automatically analyzed with ThermoHuman^®^ software version 2.12 (PEMA THERMO GROUP S.L., Madrid, Spain), a validated system [30,31] that has been used in other studies with human population [32,33,34]. The software provides mean Tsk and standard deviation values and the number of pixels, which are automatically quantified in 48 ROI for the upper body and 36 ROI for the lower body. Those initial values were integrated, considering the average Tsk values and the corresponding number of pixels of each ROI, into seven groups (see Figure 1): Whole body (Tsk_Global_): considering the 84 ROIs; trunk: considering 10 ROIs from the anterior view (Tsk_TrunkANT_) and 10 ROIs from the posterior view (Tsk_TrunkPOST_); arms: considering 12 ROIs of both arms from the anterior view (Tsk_ArmsANT_) and 12 ROIs from the posterior view (Tsk_ArmsPOST_); and legs: considering 16 ROIs of both lower limbs from the anterior view (Tsk_LegsANT_) and 16 ROIs from the posterior view (Tsk_LegsPOST_). The ROIs were integrated with the use of the equation: Tsk_integrated_ = (Tsk_ROI_^1^ × npix_ROI_^1^ + Tsk_ROI_^2^ × npix_ROI_^2^ + …+ Tsk_ROI_^n^ × npix_ROI_^n^)/(npix_ROI_^1^ + npix_ROI_^2^ + … + npix_ROI_^n^), where “n” is the number of ROI to be integrated, and “npix” is number of pixels included in the ROI. The data of the head, hands, gluteus, hips, and feet were excluded from the analysis.

#### 2.2.3. Statistical Analysis

The Kolmogorov–Smirnov test was applied to confirm the normality of the dependent variables. As the normality was confirmed, the results are presented as average, minimum, and maximum values and their standard deviations. A Student’s *t*-test for independent samples was run to verify whether TS, CS, and Tsk differed between groups (obese and non-obese). Moreover, Cohen’s test was used to assess the effect size, which was interpreted following the scale proposed by Sawilowsky [35], which classifies the values of d as very small (0.01), small (0.2), medium (0.5), large (0.8), very large (1.2), and huge (2.0). The correlation between these variables was analyzed using the Pearson correlation test.

Furthermore, we elaborated a normative table to establish the thermal profile of the adolescents based on the %BF for each ROI analyzed. For this, we used the percentiles (P) as a reference to classify if an ROI was “strongly hypo-radiant” (P < 5), “hypo-radiant” (P < 25), in “thermal normality state” (P = 50), “hyper-radiant” (P > 75), or “strongly hyper-radiant” (P > 95). The choice of terms for characterizing the ROI was based on other studies [36,37].

The statistical analyzes were carried out by statistical software (SPSS, version 22.0), with a significance level of 5%.

## 3. Results

Table 1 presents the data on the quantity of fatness of the two participant groups (*n* = 100) based on their classification of obesity.

No differences were observed (*p* > 0.05 and 95% CI = −0.122/0.482) in the values reported for TS and CS by obese (TS = 1.01 ± 0.40 and CS = 1.53 ± 0.58) and non-obese (TS = 0.83 ± 1.00 and CS = 1.27 ± 0.95) individuals in the thermographic collection environment.

Table 2 presents the results obtained by the thermographic evaluation of the two participant groups (*n* = 100) and their respective means, standard deviation, and minimum and maximum values as well as a comparison between the values observed in the participants with and without obesity. The main Tsk differences were observed for the Tsk_Global_ (0.91 °C), Tsk_TrunkANT_ (1.28 °C), and Tsk_TrunkPOST_ (1.18 °C), being lower in obese individuals with “very large” effect size values.

This pattern of negative variation observed between the Tsk values of obese and non-obese adolescents was also verified in the correlation between the variables. We found a negative relationship between %BF_global_ and Tsk_global_ (r = −0.57, *p* < 0.001), between the %BF_Trunk_ and Tsk_TrunkANT_ (r = −0.71, *p* < 0.001) and Tsk_TrunkPOST_ (r = −0.65, *p* < 0.001), %BF_Arms_ and Tsk_ArmsANT_ (r = −0.29, *p* < 0.01) and Tsk_ArmsPOST_ (r = −0.36, *p* < 0.001), and %BF_Legs_ and Tsk_LegsANT_ (r = −0.45, *p* < 0.001) and Tsk_LegsPOST_ (r = −0.44, *p* < 0.001), with emphasis on the values observed in the trunk region, as illustrated in Figure 2.

Based on the results obtained, Table 3 suggests breakpoint values to classify the person (both obese or non-obese) according to their level of infrared radiation as “strongly hypo-radiant” (P < 5), “hypo-radiant” (P < 25), “in thermal normality state” (P = 50), “hyper-radiant” (P > 75), or “strongly hyper-radiant” (P > 95) on all the considered integrated ROIs.

## 4. Discussion

The main results observed in this study suggest that the Tsk of individuals considered obese is lower than those without obesity (Table 2). Among the results, we highlight the effect size values observed in the evaluations of the Tsk_GLOBAL_, Tsk_TrunkANT_, and Tsk_TrunkPOST_, which presented “d” values of 1.23, 1.64, and 1.57, respectively, representing a probability of 80.8%, 87.6%, and 86.7% for an obese adolescent presenting lower Tsk values than a non-obese adolescent for these ROIs. Additionally, Tsk values are inversely related to %BF for all ROIs analyzed in the study, highlighting the results observed between %BF_global_ and Tsk_global_ (r = −0.57, *p* < 0.001), %BF_Trunk_ and Tsk_TrunkANT_ (r = −0.71, *p* < 0.001), and %BF_Trunk_ and Tsk_TrunkPOST_ (r = −0.65, *p* < 0.001). These data make it possible to affirm that this parameter should be considered in studies evaluating Tsk by IRT once the range of thermal normality varies according to the obesity classification of the evaluated patient. For this reason, we propose tables for the characterization of thermal normality to minimize any error in evaluation of the thermal images according to the classification of obesity for male adolescents.

The influence of %BF on Tsk values assessed by IRT has already been verified in other studies with the adult population based on different analysis models and presenting similar results to the present study. Chudecka et al. [22] and Chudecka and Lubkowska [23] used the bioimpedance technique and manual marking of ROIs to assess the impact of %BF on Tsk in adult women. Chudecka et al. [22] compared 20 obese women (23.2 ± 1.57 years, 90.7 ± 5.12 kg, 167.2 ± 3.75 cm height, and 37.8 ± 2.25 %BF) with 20 non-obese women (22.4 ± 1.22 years, 60.4 ± 2.56 kg, 169.0 ± 2.68 cm height, and 25.7 ± 2.44 %BF), verifying that women with obesity presented lower values (*p* < 0.05) of Tsk in the anterior and posterior regions of the arms, thighs and calves, the abdomen, and lower portion of ribs. In addition, they presented a negative correlation with %BF on the anterior (r = −0.77, *p* = 0.001) and posterior (r = −0.63, *p* = 0.008) regions of the thigh and abdomen (r = −0.88, *p* = 0.000). The body fat of the abdomen region was also negatively correlated (r = −0.59, *p* = 0.052) with Tsk in the study by Chudecka and Lubkowska [23], who compared 15 women with anorexia nervosa (18–24 years, 44.9 ± 4.49 kg, 169.90 ± 6.16 cm of height, and 13.30 ± 1.43 %BF) with 100 apparently healthy women (21–23 years old, 62.0 ± 4.84 kg, 168.8 ± 6.12 cm of height, and 22.8 ± 3.77 %BF). In both situations, the women stayed 20 min at a room temperature of 25.0°C and 60% relative humidity before imaging.

Neves et al. [17] and Salamunes et al. [21] used DXA for analyzing body composition of an adult population including both men and women, and the impact of %BF on the observed Tsk values also presented results equivalent to those of the present study. In the study by Neves et al. [17] that evaluated the Tsk in 47 men and 47 women aged between 18 and 28 years, after 15 min at a room temperature of 23.0 ± 1 °C (no mention of humidity), they observed that the highest value of %BF was negatively correlated with the average Tsk of the anterior (r =−0.76, *p* < 0.05) and posterior trunk (r = −0.69, *p* < 0.05), anterior (r = −0.57, *p* < 0.05) and posterior lower limbs (r = −0.63, *p* < 0.05), and anterior (r = −0.42, *p* < 0.05) and posterior arms (r = −0.47, *p* < 0.05) in males and also negatively correlated with the anterior (r = −0.27, *p*< 0.05) and posterior trunk (r = −0.47, *p* < 0.05), anterior (r = −0.36, *p* < 0.05) and posterior lower limbs (r = −0.40, *p* < 0.05), and anterior (r = −0.30, *p* < 0.05) and posterior arms (r = −0.21, *p* < 0.05) in women [18]. This negative correlation in women was also reported by Salamunes et al. [21], who evaluated 123 women aged between 18–35 years after 15 min at a room temperature of 21.0 °C (no mention of humidity), observing this behavior in the anterior and posterior regions of the trunk (r = −0.33 and r = −0.36, *p* = 0.000, respectively), anterior and posterior arms (r = −0.40 and r = −0.43, *p* = 0.000, respectively), and anterior and posterior lower limbs (r = −0.38 and r = −0.49, *p* = 0.000, respectively).

The results in the present study, corroborated by those who observed the same Tsk pattern and its relation with the %BF, clearly demonstrate that the adipose tissue influences the Tsk values, probably due to its low thermal conductivity [10,11]. Thus, taking body fat into account is important when analyzing thermographic images. For this reason, we present values for the characterization of thermal normality according to the subject’s obesity classification (Table 3). We propose the points of thermal normality (P = 50) and cutoff points of P < 25 for low radiating and P > 75 for high radiating ROIs and cutoff points of (P < 5) for “very low” and (>95) for “very high” radiating ROIs. This proposal is very innovative, and it has not been conducted by other studies that evaluated the thermal profile in adults [18,38,39,40,41,42] or that observed differences between Tsk values as a function of body composition [17,21,22,23] or anthropometric indexes [22,23].

To the best of our knowledge, this is the first study that evaluated the impact of %BF on Tsk values in adolescents using DXA to estimate body composition and presents a different analysis methodology from previous studies, proposing a table of thermal normality. While previous studies used manual marking methods for ROIs selection, this study used software with automatic selection that has already been used in other studies for thermographic evaluation [5,43]. This characteristic can reduce individual error and promote greater reliability of the data obtained.

Our results can contribute to the process of thermographic evaluations, providing a new understanding of previous studies that sought to understand the population Tsk profile [38,39,40,41,42] without taking the %BF into account, which can lead to a misevaluation of the characteristics of the evaluated individuals and may cause an erroneous diagnostic action. Therefore, it is important that future studies that aim to draw a population’s thermal profile carry out their characterization in terms of %BF or anthropometric indexes related to this variable. In order to allow a better understanding of thermal images, a possible suggestion is to stratify different %BF classification ranges to establish more specific normality values for differences in body fat. Despite being considered the reference method for assessing %BF, DXA is an expensive technology with limited accessibility. In this way, researching the influence of body composition on Tsk, the body mass index (BMI) appears as a viable option; however, it requires a specific evaluation since different BMI classification ranges can also influence Tsk values in adolescents, as indicated by Reis et al. [34,44]. However, it is important to observe whether the subject performs resistance training activities since the total amount of muscle mass can influence the BMI. We emphasize that it was considered an inclusion criterion in this study to refrain from performing resistance training.

The observed results demonstrate that the Tsk values considered normal for individuals considered obese are different from those considered non-obese. Thus, male adolescents evaluated by IRT in search of diagnostic help on some muscle group pain should be framed in their respective body composition range to avoid general errors on the part of the clinical staff, for example. In addition, knowing this relationship can also influence an evaluation in sports where it is common for players to start the pre-season with higher body fat values or in sports that can be categorized by body weight, where it is normal to find different BF% and BMI patterns. Given the subject’s characteristics, understanding how and to what extent this factor can influence the IRT helps in decision making and in the evaluation process, mainly when the professional assembles a thermographic mapping of the subject throughout the season. Another possibility is to check for pathological skin changes ranging from malignancies (e.g., melanomas) and autoimmune disorders (e.g., atopic dermatitis or AD) to infectious conditions such as herpes simplex, which also lead to unique types of changes.

Since one of the limitations of the study was that it was only carried out with Brazilian male adolescents aged 16.79 ± 0.97 years, we suggest performing similar studies with different genders and age groups; for example, women, who tend to have greater %BF and may suffer more considerable alterations in Tsk values for that reason, or the elderly, who suffer orthopedic, metabolic, and thermoregulatory disturbances. Thus, these two population groups can benefit considerably from this strategy, allowing better evaluation of the resulting images and allowing the professional to understand whether the evaluated region is “hypo-radiant”, in a “thermal normality state”, or “hyper-radiant” depending on the clinical context that the patient is undergoing. In addition, we suggest conducting similar studies at different temperature and humidity ranges in the thermographic collection room to verify whether this can influence the results. We emphasize that the study was carried out within established standards for thermographic collection, and it was demonstrated that the participants felt thermally comfortable subjected to the temperature and humidity of the room. We also suggest that in future studies, the activity of the sympathetic neuro vegetative system be controlled since it may influence the measurement in comparative cases, as it was in the present study, by promoting changes in blood flow. It is essential to improve the application of the technique continuously.

It is important to highlight that the procedures for obtaining thermographic images in this study followed specific guidelines related to the collecting device. However, as the evaluation of IRT in humans is in constant technological evolution, we also suggest that other evaluations investigate whether different thermographic cameras (mainly with better resolution and precision) observe the same pattern of results presented in the present study.

Understanding the factors that can influence the Tsk values obtained by IRT is crucial for evaluating thermographic images to be used as an auxiliary tool in the diagnosis of the alterations in the individual’s normality pattern. Regarding the present study, the results make it clear that the %BF is a variable that must be considered in the thermographic image analysis, which can improve the use of IRT in clinical and sports environments and/or in the physical rehabilitation process.

## 5. Conclusions

Adolescents with a higher amount of body fat had lower Tsk values, with a negative correlation shown between them and influencing the evaluation of the thermographic image, which should be carefully observed. Normality classification values for Tsk were proposed according to the evaluated classification—with or without obesity—which can be used as a reference in the evaluation of thermographic images obtained in a collection environment similar to the one in the present study.

## Figures and Tables

**Figure 1 ijerph-20-04405-f001:**
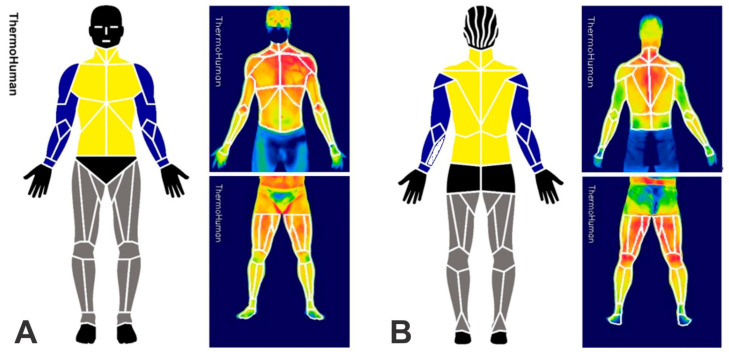
ROI analyzed by ThermoHuman^®^ in anterior (**A**) and posterior (**B**) views and an example of evaluation by the software. Note: arms, blue; trunk, yellow; legs, gray.

**Figure 2 ijerph-20-04405-f002:**
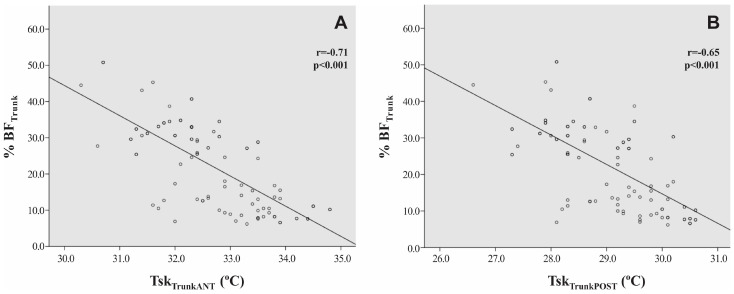
Correlation between values for %BT_Trunk_ and Tsk_TrunkANT_ (**A**) and Tsk_TrunkPOST_ (**B**) (*n* = 100).

**Table 1 ijerph-20-04405-t001:** Body fat values (in %) obtained by evaluating with DXA and comparing according to the classification of obesity in adolescents (*n* = 100).

Area (%BF)	Obese (*n* = 50)			Non-Obese (*n* = 50)			*p*
	Mean ± SD	Min.	Max.	Mean ± SD	Min.	Max.	
Total	30.21 ± 3.79	25.00	47.50	11.33 ± 3.08	5.90	22.50	<0.001
Trunk	32.57 ± 4.46	24.60	50.80	11.73 ± 3.47	6.20	25.90	<0.001
Upper Limbs (Arms)	23.22 ± 4.61	15.00	41.10	7.14 ± 2.16	4.10	15.90	<0.001
Lower Limbs (Legs)	31.46 ± 4.25	23.40	49.20	12.73 ± 3.39	6.10	23.30	<0.001

Note: SD, standard deviation; Min., minimum values; Max., maximum values.

**Table 2 ijerph-20-04405-t002:** Observed values and Tsk differences between the obese (*n* = 50) and non-obese (*n* = 50) groups.

ROI (°C)	Obese	Non-Obese	*p*	Difference Mean ± SD (95% CI)	Effect Size (d) (95% CI)
	Mean ± SD (Min/Max)	Mean ± SD (Min/Max)			
T_skGLOBAL_	30.83 ± 0.66 (29.11/31.87)	31.74 ± 0.81 (29.36/33.25)	<0.001	0.91 ± 0.15 (0.61/1.20)	1.23 (0.80/1.66)
Tsk_TrunkANT_	32.04 ± 0.74 (30.32/33.47)	33.31 ± 0.81 (31.63/34.75)	<0.001	1.28 ± 0.1 (0.97/1.59)	1.64 (1.18/2.09)
Tsk_ArmsANT_	31.80 ± 0.70 (30.45/33.14)	32.20 ± 0.84 (30.24/33.96)	<0.05	0.40 ± 0.16 (0.09/0.71)	0.52 (0.12/0.92)
Tsk_LegsANT_	31.33 ± 0.75 (29.27/32.67)	32.10 ± 1.16 (29.13/34.26)	<0.001	0.78 ± 0.20 (0.39/1.16)	0.79 (0.60/1.43)
Tsk_TrunkPOST_	28.48 ± 0.78 (26.65/30.16)	29.66 ± 0.72 (28.14/30.61)	<0.001	1.18 ± 0.15 (0.89/1.47)	1.57 (0.77/1.62)
Tsk_ArmsPOST_	30.55 ± 0.80 (28.98/32.47)	31.16 ± 0.99 (28.23/33.16)	<0.001	0.61 ± 0.18 (0.25/0.96)	0.68 (0.27/1.08)
Tsk_LegsPOST_	30.14 ± 0.84 (27.55/31.99)	30.86 ± 1.00 (27.79/32.53)	<0.001	0.72 ± 0.18 (0.35/1.09)	0.78 (0.37/1.19)

Note: SD, standard deviation; Min., minimum values; Max., maximum values.

**Table 3 ijerph-20-04405-t003:** Percentile intervals for Tsk according to the considered integrated ROIs and %BF classification.

RCIs (°C)	%BF Classification									
	Obese (*n* = 50)					Non-Obese (*n* = 50)				
	P_5_	P_25_	P_50_	P_75_	P_95_	P_5_	P_25_	P_50_	P_75_	P_95_
T_skGLOBAL_	29.75	30.58	30.85	31.38	31.73	30.54	31.20	31.78	32.30	33.04
Tsk_TrunkANT_	30.66	31.51	32.09	32.41	33.29	31.90	32.78	33.46	33.80	34.54
Tsk_ArmsANT_	30.58	31.28	31.74	32.30	32.82	30.64	31.84	32.26	32.60	33.57
Tsk_LegsANT_	30.02	30.81	31.33	31.87	32.37	30.47	31.29	32.14	32.91	33.91
Tsk_TrunkPOST_	27.29	27.95	28.38	29.15	29.49	28.26	29.21	29.78	30.17	30.55
Tsk_ArmsPOST_	29.26	29.90	30.56	31.24	31.68	29.69	30.68	31.03	31.89	32.69
Tsk_LegsPOST_	28.64	29.65	30.16	30.78	30.95	29.50	30.18	30.92	31.64	32.30

## Data Availability

The data that support the findings of this study are available from the corresponding author, H.H.T.R., upon reasonable request.

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
