# Peer review of "Can Adipose Tissue Influence the Evaluation of Thermographic Images in Adolescents?"

_ijerph, 2023, doi:10.3390/ijerph20054405_

Round 1

Reviewer 1 Report

1.       Line 40. Rewrite, authors are giving undue authorship credit. The cited reference is a generic review, and the cited author is not the one who originally suggested the factors affecting the analysis, but other authors who preceded him. There are specific guidelines for thermographic evaluation that have already defined the minimum necessary to obtain quality images that can be used for evaluation and interpretation of results.

2.       From a medical imaging point of view the authors are mixing “image quality” with “image interpretation” (l.43 and 55). They are different radiological concepts. In any method of medical diagnostic imaging, it is assumed that quality must always be assured for correct interpretation. Without quality, there is no medical interpretation. It made no sense in the text.

3.       Therefore, when referring to body composition, this is related to “image interpretation” and not “image quality”. The quoted reference also brings this confusion in its original text and the error is propagated in this study. Regardless of body composition, “image quality” must always be achieved by the technician who captures the image, be it thermography or any other imaging diagnostic method. It is an obligation of good technique that is independent of body composition. If that were the case, it would not be possible to analyze obese or cachectic patients. Even having different images due to different metabolic behavior, they must have good “image quality” to be interpreted correctly. Conceptually revise the text (in the discussion as well).

4.       In line 52 the text is not clear, it is generic and does not specify by body region (as in hypothesis l.64). Obese individuals may have higher hand temperature than non-obese patients. Non-obese individuals may have a lower nasal temperature than obese patients, eg. Which body regions specifically is this relationship mentioned in the text referring to? Rewrite the introduction and hypothesis, and clarify basic concepts of human thermology. The most accurate evaluation of the images is not simply the Tsk measurement, as well as the radiopacity of the radiological image, but the distribution pattern (in this case, thermal signature) and the comparative clinical context to be able to “interpret” a medical image.

5.       How much does the thermal conductivity of fat influence skin temperature if there are perforating vessels that reach the dermis? How much is this relationship? Contextualize.

6.       Does the sympathetic neuro vegetative system influence the vasomotor control of cutaneous blood flow how was this controlled in the study so that patients could be compared to each other?

7.       In the third paragraph there is a series of citations of articles that create confusing and erroneous reasoning on the subject. In the text, the authors are considering subcutaneous fat as a thermal insulator as if it did not have perforating cutaneous blood vessels that reach and perfuse the skin. If these blood vessels were blocked by fat this would be the same as saying that the skin of obese patients is more ischemic than thin people. Therefore, it is not a true physiological statement. Check bibliographical references about it.

8.       In l.55 the authors are mixing radiological medical concepts of “precision” and “efficiency”. “Accuracy” and “efficiency” are independent of body composition, as mentioned above. Review the imaging concept and rewrite. It is already based on the premise that professionals who work with IRT should use clinical evaluation equipment previously validated at the factory in terms of its accuracy, ie, to be specific to be used in the evaluation of the human body. When a diagnostic ultrasound or tomography device is used, for example, they must be accurate and efficient, regardless of the patient's body composition, for the professionals who will work with the interpretation of these images. Surprisingly, in the text that is concerned with precision, the authors used a device with an accuracy of only 2%, which is a very low accuracy for clinical evaluation compared with other medical devices such as ultrasound, tomography, and MRI. This may be related to the fact that these sensors are calibrated for very high non-physiological temperatures, ie, above 100º C or below 10º C.

9.       In line 62, make it clearer what type of diagnosis the authors are referring to: nosological, anatomical, functional, anatomopathological, structural, initial, early, evolutionary, screening, triage, etc. The concept is very broad.

10.   Explain what is “noise disturbances” in l.83.

11.   The authors separated individuals based on BMI. But did they not consider that patients with a higher BMI could have a higher percentage of lean mass and be mistakenly classified as obese? How was this assessed and confirmed?

12.   The evaluation was conducted at 21º C, ie, ambient temperature below the physiological limit. In tropical countries, guidelines recommend thermalization at a temperature of 23º C. According to vascular medical studies, this difference of 2º C can significantly influence cutaneous blood flow by causing an accentuated supraphysiological cold stimulus that precedes shivering. A room temperature of 21º C is used for the evaluation of inflammatory states in arthritis studies, and according to the literature, it may interfere with the evaluation of neurological disorders. Therefore, the recommended temperature is 23º C for the study of normality tables in tropical countries.

13.   On line 114, the authors use the term acclimatization, often erroneously used in articles. The correct concept would be thermalization, ie, a shorter period of “time” in which the patient is exposed to a certain ambient temperature. Acclimatization refers to long periods when the body adapts to a particular “climate”. “Climate” and “time adaptation” are completely different concepts in thermology. The classic guidelines consider 15 minutes as the ideal time for thermalization, especially when patients are not completely exposed, ie, if they wear slippers and shorts.

14.   Although the term “hyperthermia” is commonly erroneously cited in the literature, the authors here cite correctly the term “hyper-radiant”, but did not inform the bibliographic reference or articles that use this same term. To check and add.

15.   Obese people have a larger body surface area than lean people, ie, the number of pixels in the ROI of obese people is greater than that of non-obese people. Mathematically, it is a fact that having a greater number of thermal points per area (pixels) the temperature tends to be lower in obese than smaller pixels per area in non-obese. How did the authors analyze this difference in the number of pixels per ROI in the presented results? Check mathematical equations that can confirm or adjust the results presented in this study.

16.   This above may also explain the results obtained by the other authors cited in the article. To check and comment.

17.   The height of the participants was not informed. Were the images captured at the same distance from the camera for all individuals, or did the shortest individuals have closer images and the tallest individuals farther away? Regardless of the distance, how was the analysis of the results presented regarding the difference in the number of pixels per ROI that varied?

18.   The fact that software brings an average temperature value automatically selected by body parts does not guarantee security in the analysis (nor does it promote greater reliability) if you do not take into account this difference mentioned above and neither the 2% accuracy of the device. This may be critical in responding to the authors' hypothesis. Therefore, increasing the quality of the interpretation of obtained data has nothing related to the software, this statement is not true. Check the text (also in the discussion). The authors are hyper estimating the software.

19.   The thermal image is two-dimensional and is evaluating the measurement of curvilinear and small structures such as the adductor region or folds in the inguinal and axillary region. It is also not considering deviations or small rotations of the body that can make a greater ROI on one side compared to the other. How is this handled in thermography? This can be a major limitation in accurately measuring temperature. How did the authors (or software) deal with this problem when analyzing their results?

20.   Since the article talks about image quality, an example of thermal images collected from one of the participants was missing along with Figure 1.

21.   Why were only 100 patients analyzed in Figure 2 and the others?

22.   Add the p values in the caption of Figure 2.

23.   Do not start the discussion with the objective of the paper, this is in the wrong part. Rewrite.

24.   At l.186 TskTrunkANT is repeated. To check.

25.    At l.187 on probability, rewrite. It is more interesting to the reader the other way around, to know whether the subject can be classified as obese or non-obese based on skin temperature. This would make more sense in the text.

26.    At l.189 “the tsk values are inversely related in the %BF” inform how much this average ratio is. Bring numerical data.

27.   What is the practical effect of having a table of temperature values in obese and non-obese young adolescents? What would this change in a clinical diagnosis? Practical examples of this “normality table” were not clear.

28.   If it was done at thermoneutral room temperature (23oC), would the results be the same? Further, describe this study's limitations.

29.   For what type of clinical situation would a temperature reference chart be useful in obese and non-obese young male adolescents? Give practical examples that justify the statement of l.258. Not it is clear also in l.268 (clinical and sports environments, physical rehabilitation process)

30.   Make clear the limitation of the study being only of a very restricted normality table only in young Brazilian males aged 16 years old.

31.   In the comparative studies cited by the authors in the discussion, were these carried out under the same conditions? That is, the same ambient temperature, humidity, and thermalization time? Cite this in the text.

32.   In l.251, define “clinical precision”. What are the authors referring to? Revise and rewrite.

33.   In l.262 the thermal image analysis process is independent of BMI, gender, or age. The term is not correct. The temperature values may be different, but the analysis depends on the clinical context. The text confuses the reader about the basic concepts of interpretation and diagnostic imaging and may devalue thermography as a diagnostic tool, as it raises concerns about the erroneous way in which the text is written. It should be more specific to what it proposes, ie, from a purely physiological and not a diagnostic point of view, auxiliary tool in the diagnosis, image analysis, or clinical interpretation which is different from a medical point of view.

34.   For those who work with diagnostic imaging, the concept that %BF or BMI must be taken into account in the interpretation is obvious, it would not be different for thermography or another diagnostic method (even laboratory blood analysis).

Reviewer 2 Report

Do you need so many citations to support those statement in the introduction (you have used 7(38 references for that).?

How did you conduct a selection process, as it is not clear how did you choose 100/216 participants? If left unexplained this may represent the selection bias regarding the obese participants. According to the text, you have randomly chosen 50 controls without obesity. Also, how did you come up with N=100 for this study, was sample size calculated a priori, and if so, how?

Citation 19 is from wrong journal it belongs to AJPH not Blood. Please re-check all citations in the text!

How did you control for humidity? Which measures would you take in case of humidity change?

A line separating obese and non-obese in Table 3 would be appreciated for easier reading.

Line 249: remove extremely

I disagree that same thing can be done using BMI instead of %BF. BMI does not tell you anything about the body composition, and your hypothesis is that Tsk changes are due to conductivity. With BMI you can not explain that.

In conclusion try to tune down from must to should. Also you can claim this only for male adolescents. There is no clear applicability of this study. Can you add some practical examples of why and when is the approach you are proposing actually useful.

Round 2

Reviewer 1 Report

Congratulations on the revisions; the text is now much more readable, and various edits and additions have been made. Nonetheless, certain issues remain to be rectified and added:

Points 2 and 3: The authors should avoid using the phrase "image interpretation" in this paper because it confuses the reader. This paper is solely concerned with the accuracy and objective measurement of adolescent thermal patterns. It has nothing to do with medical or clinical interpretation. Try it on different papers. On line 390, the authors cannot state or conclude that the findings can influence thermal image interpretation because it was not the purpose of this study to evaluate image interpretation. Correct the phrase. Arrange in line 93 of the hypothesis as well (Point 4 ).

The summary (line 31) also confounds the final statement that %BF should be taken into account for a more precise IRT evaluation in adolescents. The word "precise" should be removed from the IRT evaluation. Because, whether obese or not, what matters is that the thermal value obtained is consistent and reliable. Obese people have specific clinical characteristics that are considered when medically interpreting the images, which is subjective and has nothing to do with the measurement's accuracy. As a result, %BF is unaffected by the medical interpretation of the image, as long as the value is accurate and obtained correctly using good technical practice. The fact that the individual is obese is an intrinsic characteristic that the evaluator considers when interpreting. This is the case when evaluating an obese person's knee in a radiographic image p.ex.

This is why, in Point 1, it is critical to give undue credit to the authors who described the classic thermography guidelines.

Point 4 is still unresolved.

Line 155, about 2% accuracy; it is important to mention this limitation because all values are influenced by the value obtained by the sensor.

Regarding Point 12, it must be stated in the conclusion that the thermal results obtained refer to a study conducted at an ambient temperature of 21o C, contrary to some authors cited in the discussion.

Point 20. You forgot to include a thermal image of the study in the diagram in Figure 1.

Reviewer 2 Report

Thank you the revised version of the manuscript. I think it can be read easier at this point, and some major potential flaws were additionally explained.
